

# Physiological parameters and differential expression analysis of *N*-phenyl-*N′*-[6-(2-chlorobenzothiazol)-yl] urea-induced callus of *Eucalyptus urophylla* × *Eucalyptus grandis*

Lejun Ouyang, Zechen Wang, Limei Li and Baoling Chen

College of Biological and Food Engineering, Guangdong University of Petrochemical Technology, Maoming, China

## ABSTRACT

In this study, we analyzed differences in the enzyme activities and transcriptomes of embryogenic and non-embryogenic calli to gain insights for improving the success of tissue culture-based breeding. A total of 2,856 differentially expressed genes (DEGs; 1,632 up-regulated and 1,224 down-regulated) were identified based on RNA sequencing and verified by reverse transcription quantitative polymerase chain reaction. Gene set enrichment analysis revealed that many of the up-regulated DEGs in embryogenic callus were enriched in the photosynthesis processes. Furthermore, the enzyme activity, hormone content, and cytokinin oxidase/dehydrogenase (*CKX*) gene expression analyses were found to be consistent with the transcriptome results. Cytokinin biosynthesis in *N*-phenyl-*N′*-[6-(2-chlorobenzothiazol)-yl] urea (PBU)-induced embryogenic callus increased owing to *CKX* repression. Measurement of endogenous hormones by high-performance liquid chromatography revealed that, compared with non-embryogenic callus, in embryogenic callus, the indole-3-acetic acid, abscisic acid and trans-zeatin riboside content had significantly higher values of 129.7, 127.8 and 78.9 ng/g, respectively. Collectively, the findings of this study will provide a foundation for elucidating the molecular mechanisms underlying embryogenic callus differentiation and can potentially contribute to developing procedures aimed at enhancing the success of callus-based plant regeneration.

## INTRODUCTION

Species of *Eucalyptus*, belonging to the Myrtaceae family of dicotyledonous plants, are among the most commonly cultivated plantation trees worldwide. The growth of *Eucalyptus* trees tends to be superior to that of other trees used for plantation, in that these species adapt well to tropical and subtropical regions, and its wood can be used for multiple purposes, including veneer, firewood, and the production of essential oil (*Pinto et al., 2010*). *Eucalyptus* is highly valued in China for its superior wood properties, rooting ability and disease resistance (*Li, Ouyang & Gan, 2015*). Plantation forestry of *Eucalyptus*

Corresponding author
Limei Li, 276667615@qq.com

*urophylla* × *Eucalyptus grandis* supplies high-quality raw material for pulp, paper, wood and energy and thereby reduces the pressures on native forests and their associated biodiversity (*Lu et al., 2010*). Nevertheless, owing to the heterozygosity of the *E. urophylla* × *E. grandis* genetic background, germplasm improvement by crossbreeding tends to be inefficient. As an alternative approach, genetic engineering of *Eucalyptus* can be used to effectively improve germplasm resources (*Girijashankar, 2011*; *Ouyang & Li, 2016*). However, for most plants, *Agrobacterium tumefaciens*-mediated transformation depends on the effectiveness of the tissue culture methods used, of which callus induction is the initial step (*Li & Luo, 2001*). In this regard, few studies have reported the successful regeneration of *E. grandis* × *E. urophylla* via callus propagation (*Ouyang et al., 2012*; *Ouyang & Li, 2016*).

Synthetic phenylurea derivatives are potent plant growth regulators that exhibit cytokinin-like activity in various culture systems (*Chung, Chen & Chang, 2007*; *Werner & Schmülling, 2009*; *Turker, Yucesan & Gurel, 2009*; *Huang et al., 2014*; *Liu et al., 2019*), among which *N*-phenyl-*N*′-[6-(2-chlorobenzothiazol)-yl] urea (PBU) was first synthesized and purified in our laboratory (*Li & Luo, 2001*). It has been demonstrated that PBU is more efficient than 6-benzyladenine (6-BA) in *Eucalyptus* callus induction (*Ouyang et al., 2012*), and, moreover, PBU-induced callus shows a higher frequency of adventitious bud induction upon transfer to adventitious bud-inducing medium (*Li, Ouyang & Gan, 2015*).

Although embryogenic callus differentiation is recognized as a key precursor to adventitious bud induction, the mechanisms underlying embryogenic callus differentiation in *Eucalyptus* have yet to be fully determined. Furthermore, the efficiency of *E. grandis* × *E. urophylla* embryogenic callus induction is known to be highly dependent on genotype, with only a few lines possessing a high capacity for callus formation. To date, certain genes and pathways have been reported to contribute to the regulation of plant callus induction, but to the best of our knowledge, the precise function of the genes involved in this process remains unknown (*Batista et al., 2018*).

In this study, we accordingly sought to examine the differences in related enzyme activities and transcriptomes of embryogenic and non-embryogenic callus based on genome-wide transcriptome sequencing and fluorescence quantitative polymerase chain reaction (qPCR) verification. The findings of this study will provide a foundation for future studies designed to further enhance the efficiency of tissue culture and transformation procedures for plant regeneration.

## MATERIALS AND METHODS

### Plant material

As explants, we used stem segments collected from clonal seedlings of *E. urophylla* × *E. grandis* grown under aseptic conditions, which were provided by the China Eucalyptus Research Center, Zhanjiang, China.

### Callus induction

For callus induction, stem segments (4–8 mm) excised from aseptically grown seedlings were inoculated on Murashige and Skoog (MS) medium supplemented with 100 mg/L of

vitamin C, 30 g/L sucrose and 7 g/L agar in addition to 19.8 µM PBU and 0.25 µM naphthalene acetic acid (NAA) for embryogenic callus induction, or MS medium supplemented with 100 mg/L vitamin C, 30 g/L sugar and 7 g/L agar in addition to 19.8 µM 6-BA and 0.25 µM NAA for non-embryogenic callus induction. The MS medium was sterilized at 120 °C for 20 min after adjusting the pH to 5.9. Vitamin C was sterilized using a 0.22-µm pore diameter membrane microfilter prior to being combined with other components. Explants were incubated at 25 ± 2 °C in the dark for 2 weeks, and then for an additional 2 weeks under a 16 h photoperiod with a light irradiance of 50 µmol·m$^{-2}$·s$^{-1}$ emitted by cool fluorescent tubes. Callus formed on MS medium was classified by color, and used to determine physiological indices, which was performed in triplicate with equal weights of fresh tissue cut from the calli subjected each treatment.

## Enzyme extraction and activity assays

One hundred-milligram (fresh weight) samples of the two callus types were ground in liquid nitrogen with the addition of two mL of extraction solution (potassium phosphate buffer, 100 mM, pH 6.5). The homogenates were centrifuged for 15 min at 12,000×$g$ and 4 °C, and the resulting supernatants were collected as enzyme extracts and maintained at 4 °C prior to being used for determinations. Superoxide dismutase (SOD) activity was estimated as described by *Giannopolitis & Ries (1977)* in a three-mL reaction mixture containing 13 mM methionine, 50 mM sodium phosphate buffer (pH 7.5), 2 µM riboflavin, 0.1 mM EDTA, 75 µM nitroblue tetrazolium, and 20 µL of enzyme extract. SOD activity was expressed as unit per min per milligram protein. One unit of SOD activity is defined as a 50% reaction inhibition compared with the control after 10 min.

Peroxidase (POD) activity was determined using a direct spectrophotometric method at 30 °C (*Hammerschmidt, Nuckles & Kuć, 1982*), and catalase (CAT) activity was determined following the spectrophotometric method described by *Jariteh et al. (2011)*.

## Plant hormone extraction and determination

Samples of the two callus types (2 g) were ground with a mortar and pestle in liquid nitrogen, followed by extraction with 80 mL methanol. Hormone activity was determined using the direct reverse phase high-performance liquid chromatography (RP-HPLC) method (*Lai & Chen, 2002*). The homogenate was extracted for 21 h at 4 °C and thereafter centrifuged at 9,500×$g$ for 30 min at 4 °C. The resulting supernatant was collected and maintained at 4 °C prior to subsequent determinations. The chromatographic separation conditions were as follows: the mobile phase was methanol and 0.01 mol·L$^{-1}$ H$_3$PO$_4$ (42:58); the flow rate was 1 mL·min$^{-1}$; the detection wavelengths were 210, 218 and 265 nm; and the injection volume was 1.5 L. The data obtained from five independent replicates were used for statistical analysis.

## Analysis of the efficiency of qPCR amplification of *CKX* expression

For both callus types, total RNA was extracted from 300 mg of fresh callus tissue, in accordance with the protocol described by *MacKenzie et al. (1997)*. The efficiency of

**Table 1 Primer sequences for *CKX* family genes.**

| Gene | Forward sequence (5′–3′) | Reverse sequence (5′–3′) |
|---|---|---|
| *Actin* | GCACCGCCAGAGAGGAAATA | GAAGCACTTCCTGTGGACGA |
| *CKXA* | TGGCAAGAGTTCGACCTTCAA | CCCCATCAATCTTTGAATTCATGC |
| *CKXB* | TGTCTGCTGTCATACCAGATGAA | GGTTGAAGATCCTCTGCCCA |
| *CKXC* | ATGGAGGAGGTTCCGTCAGA | TGGATCTATTCACTAGCGTCCG |
| *CKXD* | CCACATTTTGGCAGTGAACGA | ACCCAGGTAAGATGGTGCAA |
| *CKXE* | CCACATTTTGGCAGTGAACGA | AACCAGGTAAGATGGTGCAA |
| *CKXF* | AGTGGGTTTGAAGACTGGCA | GGTTGAAGATCCTCTGTCCAGG |

real-time PCR amplification of cytokinin oxidase/dehydrogenase (*CKX*) genes was analyzed following the protocol described by *Schmittgen et al. (2004)*. The gene name and sequences of the real-time PCR primers used in this study are listed in Table 1.

## RNA sequencing

The total RNA obtained from the two types of callus was isolated using TRIzol reagent (Thermo Fisher Scientific, Waltham, MA, USA), with the quality and quantity of the isolated RNA being determined using a NanoDrop spectrophotometer (Thermo Fisher Scientific, Waltham, MA, USA) and agarose gel electrophoresis, respectively. First-strand cDNA was synthesized from the isolated RNA using a Maxima First Strand cDNA Synthesis Kit (Thermo Fisher Scientific, Waltham, MA, USA), and double-stranded cDNA was subsequently synthesized and amplified using random primers to obtain the final cDNA libraries. The libraries thus generated were sequenced using the Illumina HiSeq™ 2500 sequencing platform.

## Bioinformatics analysis of the RNA-seq data

Low-quality and adapter-containing reads were removed to obtain clean reads, which were then aligned to the *E. grandis* reference genome using the alignment software HISAT (*Kim, Langmead & Salzberg, 2015*). We subsequently performed transcript assembly and expression calculation using StringTie (*Pertea et al., 2016*). On the basis of alignment, transcript abundance was estimated by generating a count matrix, normalized by the total count of each library to obtain count per million (CPM) values. Transcripts with low expression (CPM < 1) and lengths below 200 bp were filtered out. Differential expression analysis was performed using the edgeR package based on thresholds of a |fold change| ≥ 2 and false discovery rate (FDR) ≤ 0.0001. Gene ontology (GO) analysis was conducted using the WEGO 2.0 database (*Ye et al., 2018*) and Kyoto Encyclopedia of Genes and Genomes (KEGG) enrichment analysis was performed using the cluster Profiler package (*Yu et al., 2012*). The RNA sequencing reads have been deposited in the NCBI database under BioProject number PRJNA541120.

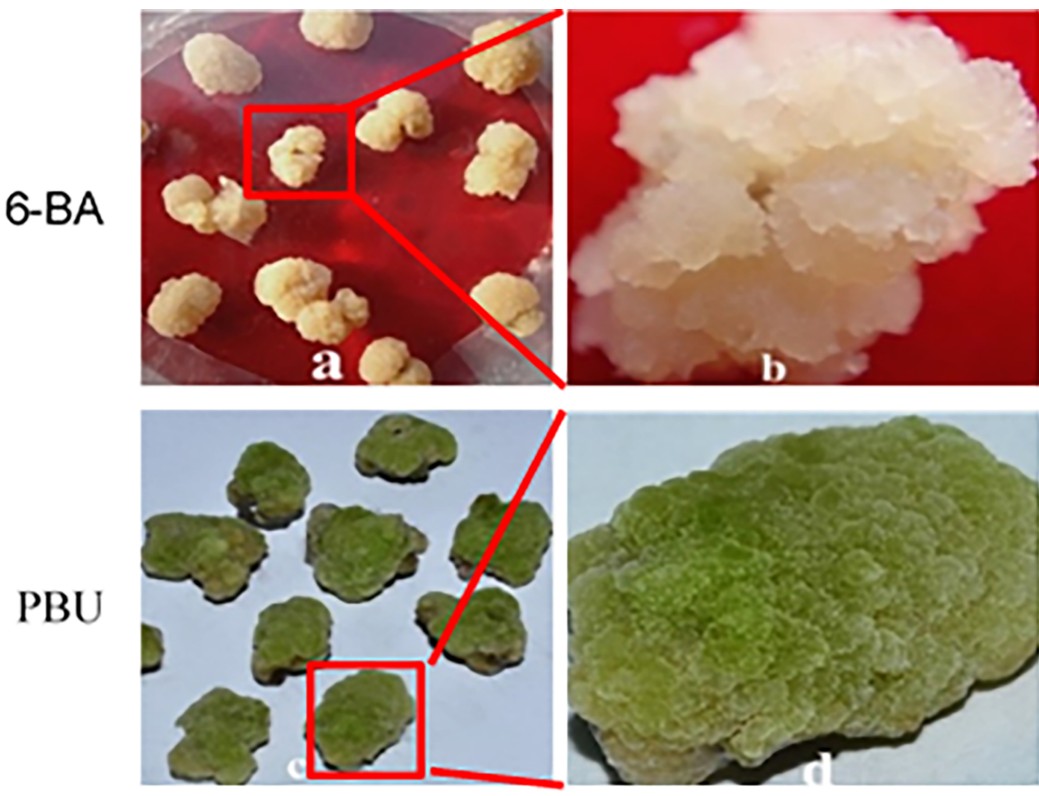

**Figure 1 Callus induced by 6-BA and PBU.** (A and B) NEC callus inoculated on MS medium supplemented with 0.25 μM NAA and 19.8 μM·L$^{-1}$ BA; (C and D) EC callus inoculated on MS medium supplemented with 0.25 μM NAA and 19.8 μM·L$^{-1}$ PBU. PBU stimulated more vigorous callus and prevented browning. In addition, callus induced by PBU showed a higher frequency of adventitious buds upon transfer to adventitious bud inducing medium.

## RESULTS

### Embryogenic callus induction using PBU

The calli that formed on MS medium were classified according to color and were used to determine selected physiological indices. Differences in the callus formation and color of embryogenic and non-embryogenic callus explants are shown in Fig. 1. The embryogenic callus were reseda and loose texture. The color of non-embryogenic callus was white, close texture. Embryogenic callus had higher vigor and is easier to induce adventitious buds than the non-embryogenic callus.

### Embryogenic callus quality assessment

Table 2 shows the four assessed physiological indices of embryogenic and non-embryogenic callus samples. We found the growth of embryogenic callus to be more vigorous than that of non-embryogenic callus. Taking regeneration potential into consideration, we hypothesized that POD activity is associated with embryogenic callus development based on the assumption that elevated concentrations of $H_2O_2$ and the accumulation of cellulose in cells are detrimental to embryogenic callus formation.

**Table 2 Differences in physiological indices between embryogenic (EC) and non-embryogenic callus (NEC).**

| Callus type | SOD activity (U·mg⁻¹ protein) | POD activity (U·mg⁻¹ protein) | CAT activity (U·mg⁻¹ protein) |
|---|---|---|---|
| EC | $228.8 \pm 5.1$** | $131.7 \pm 2.5$** | $278.1 \pm 2.5$** |
| NEC | $111.6 \pm 1.2$ | $599.4 \pm 5.2$ | $152.5 \pm 2.4$ |

Notes:
** Very significant difference ($\alpha < 0.01$).
SOD, superoxide dismutase; POD, peroxidase activity; CAT, catalase activity.

## Transcriptome sequencing and expression analysis of embryogenic callus

A total of 44,256,994 and 47,888,468 clean reads were generated for non-embryogenic and embryogenic callus samples, respectively (Table S1). Alignment of clean reads to the *E. grandis* reference genome yielded respective mapping rates of 79.4% and 69.5%, indicating that in both cases, a high proportion of reads were mapped to the reference genes (Fig. S1).

A total of 22,892 expressed genes met the criteria for further analysis. On the basis of a pairwise comparison between the embryogenic callus (case) and non-embryogenic callus (control) at thresholds of $|\log_2 FC| > 1$ and FDR < 0.0001 (Fig. 2), we identified a total of 2,856 differentially expressed genes (DEGs) (Fig. S2). Among these, 1,632 and 1,224 genes were significantly up- and down-regulated, respectively (Fig. S2), implying that these genes might be involved in PBU-induced embryogenesis.

## Functional analysis and enrichment of DEGs

To examine the relevance of the identified DEGs to embryogenesis, we initially carried out GO annotation of the expressed genes, the results of which are shown in Fig. 3.

The up-regulated DEGs were found to be mainly associated with photosynthesis, binding, oxidoreductase activity and carbohydrate metabolic processes (Fig. S3), whereas the down-regulated DEGs were enriched in ADP binding, signal transduction, and microtubule-related processes (Fig. S4). These data accordingly indicate that embryogenesis is associated with a high level of metabolic activity. We further performed KEGG enrichment analysis to determine the key pathway involved in embryogenesis, and accordingly observed predominant enrichment of up-regulated DEGs in photosynthesis, phenylpropanoid biosynthesis, zeatin biosynthesis, and glucose and tyrosine related metabolism (Fig. S5). Down-regulated DEGs were primarily mapped to plant–pathogen interaction, MAPK signaling pathway, and cutin, suberin and wax biosynthesis (Fig. S6), thereby indicating that the non-embryogenic callus had been subjected to adverse stress. Interestingly, however, we found that phenylpropanoid biosynthesis was enriched in both up- and down-regulated DEGs.

## PBU promotes cytokinin biosynthesis

The detected changes in gene families and verification of changes in *CKX* expression levels using reverse transcription (RT)-qPCR are shown in Figs. 4 and 5, respectively.

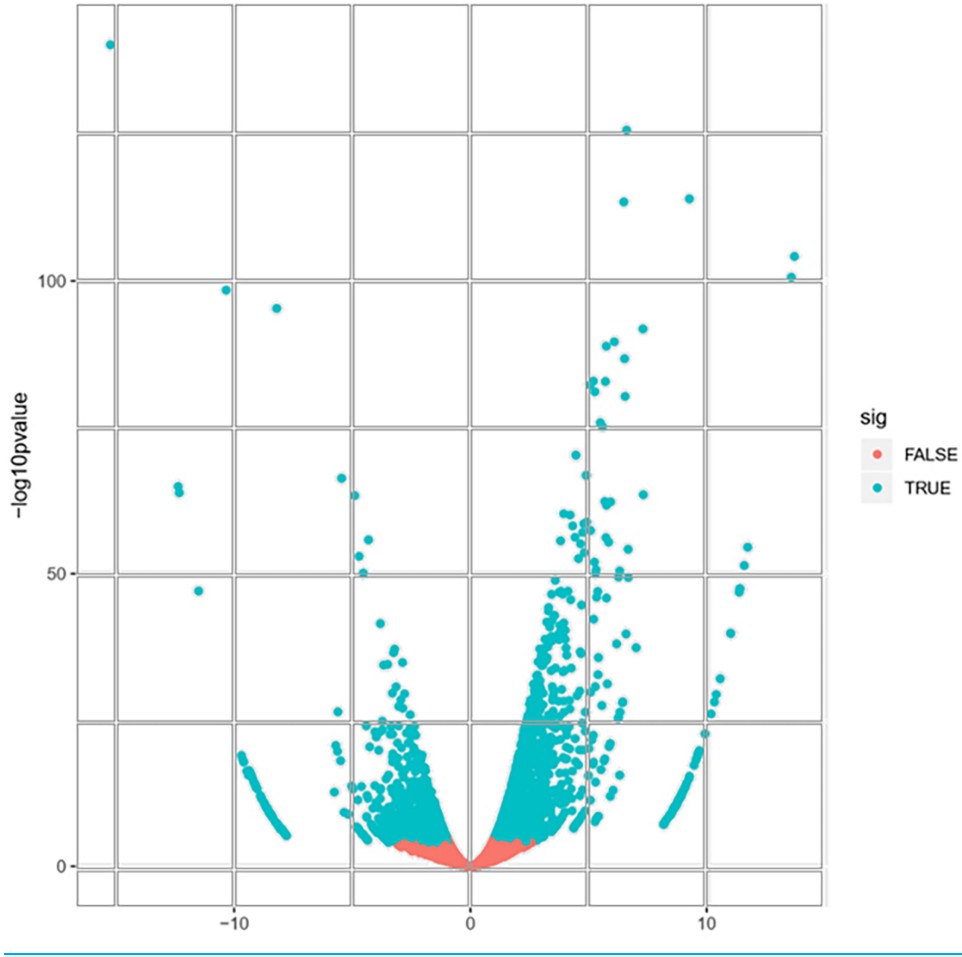

**Figure 2 Volcano plot of expressed genes.**

We believe that cytokinin biosynthesis in PBU-induced embryogenic callus was up-regulated in response to the repression of *CKX* genes, thereby indicating that PBU might promote the synthesis of cytokinins.

## Differences in plant hormones between embryogenic and non-embryogenic calli

Differences in the levels of plant hormone between embryogenic and non-embryogenic calli are shown in Fig. 6. Among the four hormone examined, we found that the contents of indole-3-acetic acid (IAA), abscisic acid (ABA), and trans-zeatin riboside (TZR) were significantly higher in the embryogenic callus than in non-embryogenic callus, whereas in contrast, the levels of gibberellic acid ($GA_3$) were considerably higher in non-embryogenic callus, indicating that $GA_3$ may reduce callus differentiation capacity and that IAA, ABA and TZR may contribute to enhancing embryogenic callus formation, including green callus induction and somatic embryogenesis.

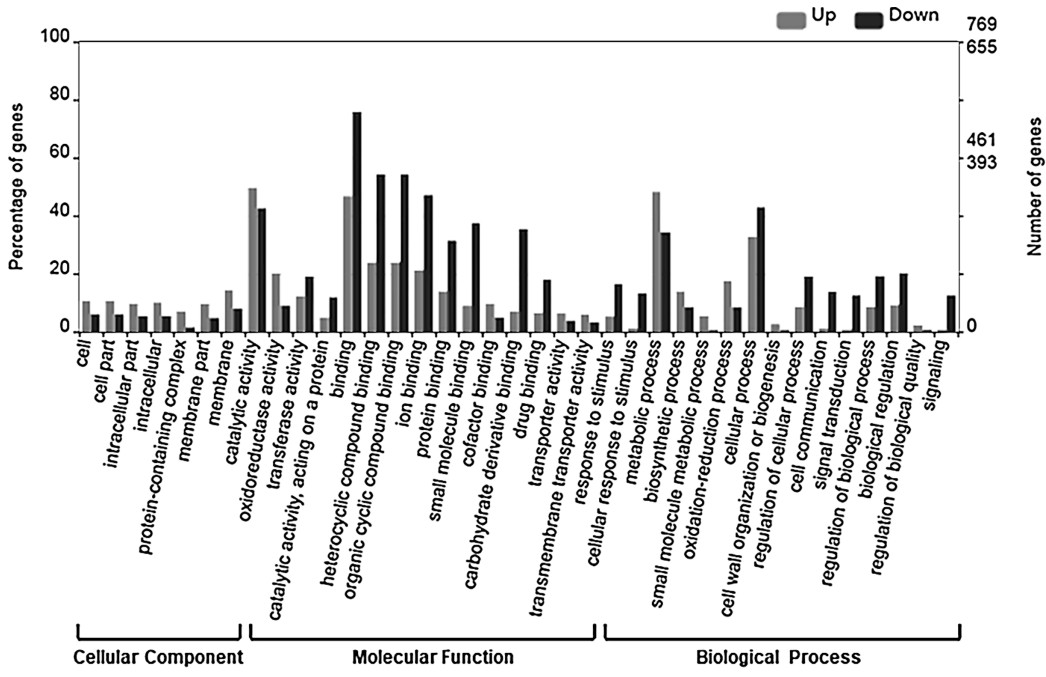

**Figure 3 GO categories of DEGs.** The DEGs were clustered into three categories, including cellular function, molecular function and biological function. Most up-regulated DEGs were mainly involved in photosynthesis, binding, oxidoreductase activity and carbohydrate metabolic processes, while down-regulated genes were mainly involved in ADP binding, signal transduction and microtubule related processes.

## DISCUSSION

Embryogenic callus induction is considered to be a key precursor to adventitious bud formation (*Hu, Xiong & Yang, 2005*; *Cairney & Pullman, 2007*), and in this regard, certain genes and pathways have been reported to contribute to the regulation of plant callus induction, with cytokinin levels being crucial for embryogenic callus establishment (*Hwang & Sheen, 2001*). In the present study, we used high-throughput sequencing to investigate changes in the transcriptomes of embryogenic and non-embryogenic calli during the processes of callus induction and establishment. On the basis of RNA sequencing analysis, we identified a total of 2,856 DEGs, among which 1,632 genes were up-regulated and 1,224 genes were down-regulated, indicating that these genes may be associated with embryogenesis. RT-qPCR analysis revealed that the observed changes in expression were highly consistent with the RNA sequencing results. Subsequent RT-qPCR analysis of *CKX* genes similarly showed the expression changes to be highly consistent with transcriptome results, thereby indicating the accuracy of the RNA sequencing. GO enrichment analysis showed that many of the up-regulated DEGs were enriched in the photosynthesis process, which is consistent with our observation of the development of green embryogenic callus indicative of vigorous photosynthesis. Furthermore, the enrichment of up-regulated DEGs significantly involved in oxidoreductase activity is consistent with changes in the enzyme activities related to oxidation-reduction reactions, including those of SOD, CAT and POD, which *Huang et al. (2014)* found to be associated

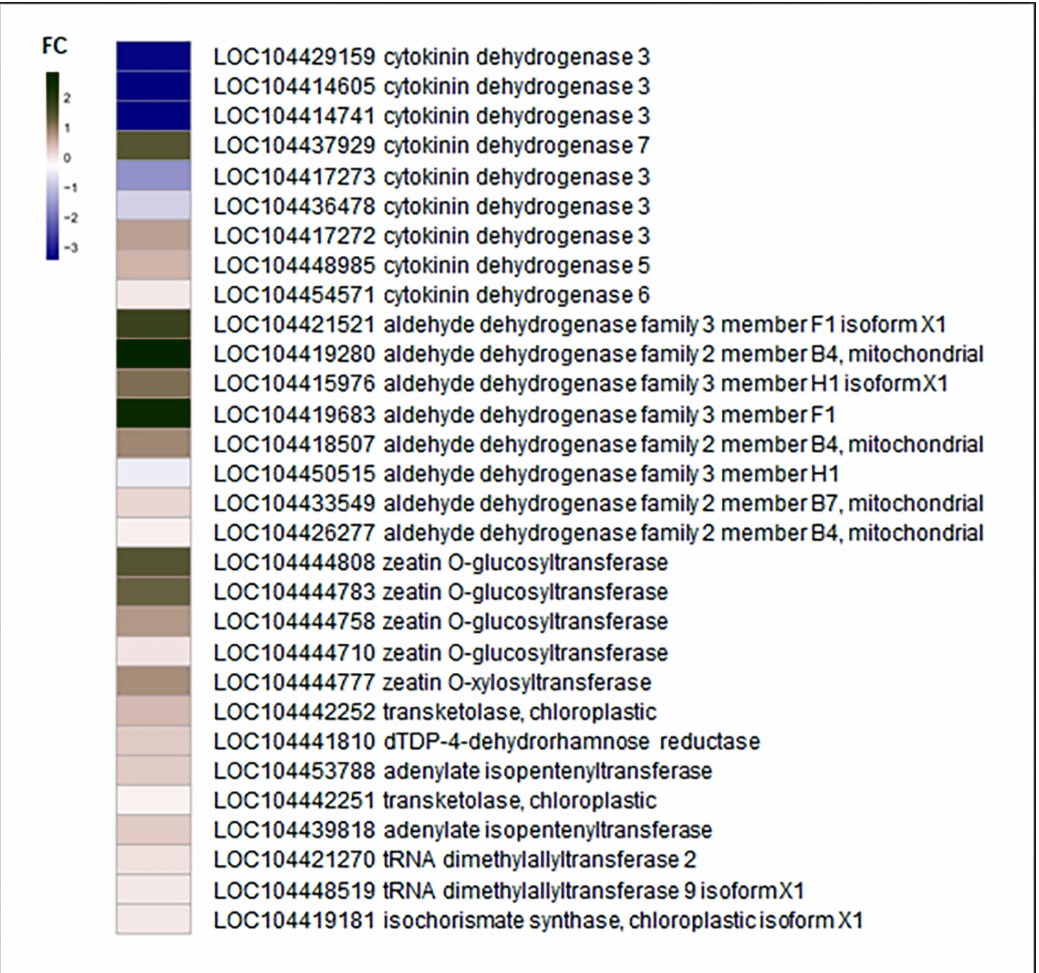

**Figure 4 Fold change of zeatin biosynthesis related genes.** Detailed expression changes of genes related to biosynthesis in EC compared to the NEC. Red and blue indicate up-regulated and down-regulated expression levels, respectively. Fold changes were calculated by CPM value. The gene name and annotation are indicted on the right.

with organogenesis. Accordingly, it can be assumed that certain concentrations of antioxidative enzymes contribute to organogenic callus formation.

Members of a multigene family encoding cytokinin oxidase/dehydrogenase proteins (*CKX*) are implicated in regulating cytokinin contents in the organs of developing plants, some of which play important roles in plant growth and development (*Zalewski et al., 2010*; *Cai et al., 2018*). Although the expression of *CKX* is generally decreased in embryogenic callus, we found that cytokinin biosynthesis was higher in PBU-induced embryogenic callus than in non-embryogenic callus, indicating that PBU might promote an increase in cytokinin levels, which is consistent with our RNA sequencing results.

Among the factors that potentially affect plant callus formation and differentiation, including minerals, growth factors, hormones, medium carbon source, and environmental factors such as temperature, light, and photoperiod, endogenous hormones are the key

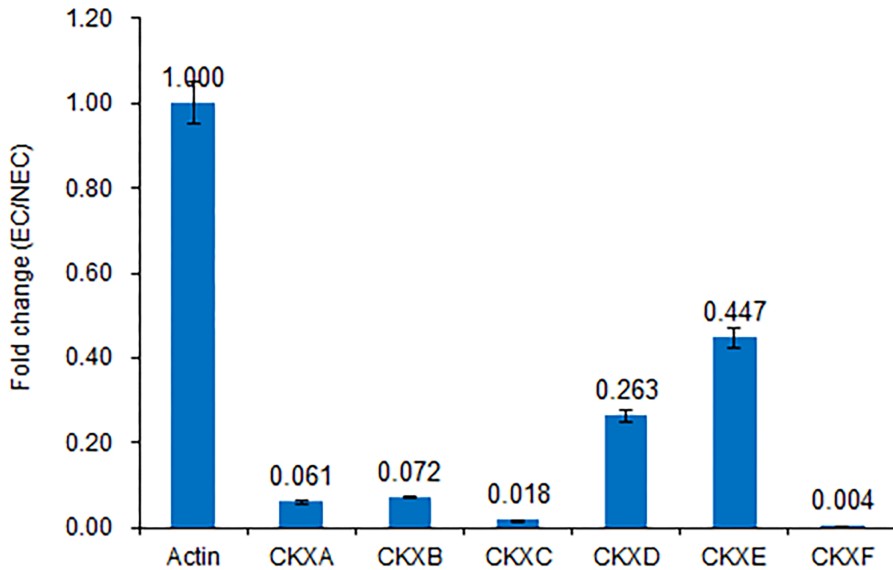

**Figure 5 Expression of *CKX* genes validated by RT-qPCR.** Consistent with the transcriptome sequencing analysis, *CXKA, CKXB, CKXC, CKXD, CKXE* and *CKXF* expression was highly decreased in the EC sample.

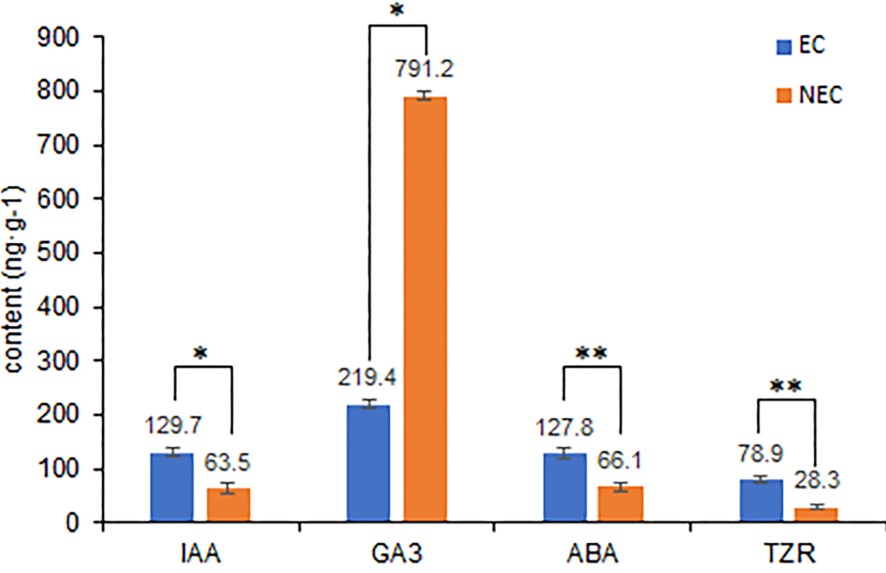

**Figure 6 Differences in plant hormone levels between EC and NEC.** Hormone content was measured by HPLC. **Indicates a very significant difference ($p < 0.01$); *Indicates a significant difference ($p < 0.05$).

regulators of developmental switch factors (*Pinto et al., 2010*). In this regard, *Prakash & Gurumurthi (2010)* observed the development of different types of *E. camaldulensis* calli in response to supplementation of the culture medium with growth regulators and obtained a higher percentage with the addition of 2 mg L$^{-1}$ ABA, which subsequently gave rise to regenerated plants. Similarly, by providing media supplemented with different hormones,

![PeerJ]

*Pinto et al. (2010)* obtained embryogenic calli of the hybrid *E. grandis × E. urophylla* showing differing characteristics. Although little is currently known regarding the role of endogenous hormones during organogenic callus formation, particularly during the primary dedifferentiation and re-differentiation associated with embryogenic callus initiation and adventitious bud differentiation, endogenous cytokinins and auxins are probably more important than exogenous factors, given that they directly determine organogenic callus progression (*Fehér et al., 2002*). Consistently, *Thomas et al. (2002)* found that sharp changes in endogenous hormone levels may be among the first important signals leading to embryogenic callus initiation, and *Zeng et al. (2007)* showed that re-differentiation is clearly correlated with a marked increase in auxin responses in cotton cells.

## CONCLUSIONS

In the present study, we obtained direct evidence for the significance of endogenous cytokinins in the expression of cellular totipotency. Our preliminary findings reveal that the embryogenic callus phenomenon is affected by environmental factors and also provide insights into the molecular mechanism of the non-embryogenic phenomenon. We believe that these findings will provide a valuable foundation for further elucidating the mechanisms underlying embryogenic callus differentiation and may also contribute to the development of procedures aimed at enhancing the success of callus-based plant regeneration.

### Funding

This research was supported by the National Natural Science Foundation of China (31470677), the Science and Technology Tackle Key Problem (2017A030303087) of Guangdong Province, the Key Project of Basic Research and Applied Research of Guangdong Province (2018KZDXM047), and the Natural Science Foundation of Guangdong Province (2019A1515010709 and 2017A030307017), and Guangdong climbing project (pdjh2019b0323). The funders had no role in study design, data collection and analysis, decision to publish, or preparation of the manuscript.

### Grant Disclosures

The following grant information was disclosed by the authors:
National Natural Science Foundation of China: 31470677.
Science and Technology Tackle Key Problem: 2017A030303087.
Key Project of Basic Research and Applied Research of Guangdong Province: 2018KZDXM047.
Natural Science Foundation of Guangdong Province: 2019A1515010709 and 2017A030307017.
Guangdong Climbing Project: pdjh2019b0323.

# PeerJ

## Competing Interests

The authors declare that they have no competing interests.

## Author Contributions

- Lejun Ouyang conceived and designed the experiments, performed the experiments, analyzed the data, prepared figures and/or tables, authored or reviewed drafts of the paper, and approved the final draft.
- Zechen Wang conceived and designed the experiments, performed the experiments, analyzed the data, prepared figures and/or tables, and approved the final draft.
- Limei Li conceived and designed the experiments, performed the experiments, analyzed the data, prepared figures and/or tables, and approved the final draft.
- Baoling Chen conceived and designed the experiments, performed the experiments, analyzed the data, prepared figures and/or tables, and approved the final draft.

## Data Availability

Data is available at GenBank: PRJNA541120.

## Supplemental Information

Supplemental information for this article can be found online at http://dx.doi.org/10.7717/peerj.8776#supplemental-information.

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
