# Peer review of "Physiological parameters and differential expression analysis of N-phenyl-N′-[6-(2-chlorobenzothiazol)-yl] urea-induced callus of Eucalyptus urophylla × Eucalyptus grandis"

_PeerJ, doi:10.7717/peerj.8776_

## Round 0.1 · original submission · Major Revisions

Dear Authors, Your manuscript needs "Major Revisions" before it is considered for publication in PeerJ. Please, address the external reviewers suggestions, comments and re-submit your "Revised Version" copy to PeerJ for further consideration.

Rev 2 has also submitted an extensively marked up manuscript for you to respond to

Thank you.

Dr. Simon Francis Shamoun,
Editor

Reviewer 1 ·

Basic reporting

.

Experimental design

.

Validity of the findings

.

Additional comments

Dear Authors,

Minor comments:

The line 47-48 something is wrong with the text, its much smaller than the rest.
Line 65 Please add space between – Embryogenic callus and (EC)
Line 120 the Authors have to add the name of the company for reverse transcriptase as for now I can see only the name for TRIZOL. In general the part of RNA sequencing have to be written in more details
Line 126 change from – Eucalyptus grandis to E. grandis, because you have already mentioned Eucalyptus before.
Lin3 137 Embryonic or embryogenic?
Line 187-199 No discussion here with the literature data at all!


Major comment

The part of the results is very short and in my opinion have to be combined with discussion. The discussion part is extremely week!

Materials and methods part have to be written in much more details.

Something wrong with the formatting! Please correct it, I can't really see some of the pictures.

Reviewer 2 ·

Basic reporting

See attached annotated copy for examples to illustrate these points:
English requires improvement in some parts. See attached annotated copy.
Insufficient references. Some relevant recent references were not included.
REferences style is inconsistent with journal requirements.
Figures and tables are adequate.
Results are explained but need expanding in some parts.

Experimental design

Improve presentation and indicate data anlaysis procedure

Validity of the findings

The finding is of scientific importance but the Discussion section is not well written. See the annotated copy.

Additional comments

I have made specific comments directly in the attached file.

Annotated reviews are not available for download in order to protect the identity of reviewers who chose to remain anonymous.

---

## Round 0.2 · Minor Revisions

Dear Authors, Your manuscript needs some minor revisions. Please, address the suggestions of the reviewer and re-submit your "Revised Version" copy to PeerJ for further consideration.

Reviewer 1 ·

Basic reporting

.

Experimental design

.

Validity of the findings

.

Additional comments

Dear Authors,

Line 155-157 This is not enough to say they you have the pictures of the calli you have to describe it. This is the results. Please describe the color, the structure of the callus etc. This is quite basic for tissue culture works. For references see:
Betekhtin et al (20016) Spatial Distribution of Selected Chemical Cell Wall Components in the Embryogenic Callus of Brachypodium distachyon
Or
Luciani et al (2006) Effects of explants and growth regulators in garlic callus formation and plant regeneration

Best wishes

Reviewer 2 ·

Basic reporting

Satisfactory

Experimental design

Satisfactory

Validity of the findings

Satisfactory

Additional comments

None

---

## Round 0.3 · accepted · Accept

Thank you for making the necessary changes on your revised version (second round review) of your manuscript.